# Oncologic Impact and Safety of Pre-Operative Radiotherapy in Localized Prostate and Bladder Cancer: A Comprehensive Review from the Cancerology Committee of the Association Française d’Urologie

**DOI:** 10.3390/cancers13236070

**Published:** 2021-12-02

**Authors:** Paul Sargos, Stéphane Supiot, Gilles Créhange, Gaëlle Fromont-Hankard, Eric Barret, Jean-Baptiste Beauval, Laurent Brureau, Charles Dariane, Gaëlle Fiard, Mathieu Gauthé, Romain Mathieu, Guilhem Roubaud, Alain Ruffion, Raphaële Renard-Penna, Yann Neuzillet, Morgan Rouprêt, Guillaume Ploussard

**Affiliations:** 1Department of Radiotherapy, Institut Bergonié, 33000 Bordeaux, France; P.Sargos@bordeaux.unicancer.fr; 2Department of Radiotherapy, Insitut de Cancérologie de l’Ouest, 44800 St-Herblain, France; Stephane.supiot@ico.unicancer.fr; 3Department of Radiotherapy, Institut Curie, 75005 Paris, France; gilles.crehange@curie.fr; 4Department of Pathology, CHRU Tours, 37000 Tours, France; gaelle.fromont-hankard@univ-tours.fr; 5Department of Urology, Institut Mutualiste Montsouris, 75014 Paris, France; eric.barret@imm.fr; 6Department of Urology, La Croix du Sud Hôpital, 31130 Quint Fonsegrives, France; beauval.jb@chu-toulouse.fr; 7Department of Urology, CHU de Pointe-à-Pitre, University of Antilles, University of Rennes, Inserm, EHESP, Irset (Institut de Recherche en Santé, Environnement et Travail)—UMR_S 1085, 97110 Pointe-à-Pitre, France; laurent.brureau@chu-guadeloupe.fr; 8Department of Urology, Hôpital Européen Georges-Pompidou, APHP, Paris—Paris University—U1151 Inserm-INEM, Necker, 75015 Paris, France; Charles.dariane@aphp.fr; 9Department of Urology, Grenoble Alpes University Hospital, Université Grenoble Alpes, CNRS, Grenoble INP, TIMC-IMAG, 38000 Grenoble, France; Gfiard@chu-grenoble.fr; 10Unité de Recherche Clinique en Économie de la Santé, CRESS METHODS INSERM UMR 1153, 75000 Paris, France; mathieu.gauthe@aphp.fr; 11Department of Urology, CHU Rennes, 35033 Rennes, France; Romain.MATHIEU@chu-rennes.fr; 12Department of Medical Oncology, Institut Bergonié, 33000 Bordeaux, France; g.roubaud@bordeaux.unicancer.fr; 13Service d’Urologie Centre Hospitalier Lyon Sud, Hospices Civils de Lyon, 69002 Lyon, France; alain.ruffion@chu-lyon.fr; 14Equipe 2, Centre d’Innovation en Cancérologie de Lyon (EA 3738 CICLY), Faculté de Médecine Lyon Sud, Université Lyon 1, 69002 Lyon, France; 15Department of Radiology, Sorbonne University, AP-HP, Pitie-Salpetriere Hospital, 75013 Paris, France; raphaele.renardpenna@aphp.fr; 16Department of Urology, Hôpital Foch, 92151 Suresnes, France; yneuzillet@afu.fr; 17Department of Urology, Sorbonne University, GRC 5 Predictive Onco-Uro, AP-HP, Pitie-Salpetriere Hospital, 75013 Paris, France; morgan.roupret@aphp.fr

**Keywords:** neoadjuvant, preoperative, radiotherapy, prostate cancer, bladder cancer

## Abstract

**Simple Summary:**

Radiotherapy may have an interesting role of reinforcing the loco-regional control of cancer, in addition to surgery, when used as a preoperative treatment. This sequence has demonstrated its efficacy and safety in various malignancies, but no strong data exist in the era of uro-oncology. In this review article, we aim to highlight the potential usefulness of preoperative radiotherapy in prostate and muscle-invasive bladder cancer, aiming to enhance pathological response and local control and to prevent intraoperative tumor seeding. We also emphasize the need for further clinical studies assessing the functional safety of subsequent surgical procedures in a competitive context of new systemic agents that have proven to demonstrate a survival benefit in locally advanced urologic cancers.

**Abstract:**

Preoperative radiotherapy (RT) is commonly used for the treatment of various malignancies, including sarcomas, rectal, and gynaecological cancers, but it is preferentially used as a competitive treatment to radical surgery in uro-oncology or as a salvage procedure in cases of local recurrence. Nevertheless, preoperative RT represents an attractive strategy to prevent from intraoperative tumor seeding in the operative field, to sterilize microscopic extension outside the organ, and to enhance the pathological and/or imaging tumor response rate. Several clinical works support this research field in uro-oncology. In this review article, we summarized the oncologic impact and safety of preoperative RT in localized prostate and muscle-invasive bladder cancer. Preliminary studies suggest that both modalities can be complementary as initial primary tumor treatments and that a pre-operative radiotherapy strategy could be beneficial in a well-defined population of patients who are at a very high-risk of local relapse. Future prospective trials are warranted to evaluate the oncologic benefit of such a combination of local treatments in addition to new life-prolonging systemic therapies, such as immunotherapy, and new generation hormone therapies. Moreover, the safety and the feasibility of salvage surgical procedures due to non-response or local recurrence after pelvic RT remain poorly evaluated in that context.

## 1. Introduction

Preoperative radiotherapy (RT) is commonly used for the treatment of various malignancies, including sarcomas, rectal, and gynaecological cancers [1,2]. Nevertheless, RT is preferentially used as a competitive treatment to radical surgery in uro-oncology or as a salvage treatment after surgery in cases of local recurrence [3,4]. RT confers oncologic equivalence for a curative purpose while preserving organ function, particularly in bladder cancer management. In that setting of RT, radical surgery such as prostatectomy (RP) or cystectomy (RC) becomes a valid treatment option for managing resistance or local recurrence after RT failure [3,4]. Nevertheless, preoperative RT represents an attractive strategy to prevent intraoperative tumor seeding in the operative field, to sterilize microscopic extension outside the organ, and to enhance the pathological and/or imaging tumor response rate [5]. Several clinical works support this research field in uro-oncology, which mainly occurs in a population of patients who are at very a high risk of local relapse.

Herein, we aimed to summarize the oncologic impact and safety of preoperative RT in localized prostate and muscle-invasive bladder cancer. We also reported the available evidence on imaging and pathological interpretation after preoperative RT as well as the outcomes after salvage surgery for both malignancies.

## 2. Methods

The primary objective was to evaluate the oncologic impact and the safety of pre-operative RT in localized prostate and muscle-invasive bladder cancer. We considered pre-operative as any RT regimen prior to radical surgery (radical cystectomy, radical prostatectomy) as part of the initial primary tumor treatment or as a salvage procedure for residual tumor or for recurrence after RT.

The secondary aims were to assess the oncologic impact of salvage surgeries after RT and the impact of pre-operative RT on imaging and pathology interpretation.

A literature web search was performed in September 2021, with no time restrictions, using AMED (Allied and Complementary Medicine), Embase, and Medline. The terms (“pre-operative”, “neoadjuvant”, “radiotherapy”) and (“radical prostatectomy”, “radical cystectomy”, “salvage”) were pooled together with the boolean operator OR. A manual search with a consultation of the references of web-search articles (authors consultation and references of web-search included articles) was performed.

Full-text articles published between 1980 and 2020 using the Roman alphabet were considered. Data extraction was conducted by three authors (P.S., S.S., G.P.). We summarized the literature evidence as follows: oncologic interest of pre-operative RT in muscle-invasive and prostate cancer; impact of RT on post-treatment imaging assessment, safety, and outcomes of surgery after RT for bladder and prostate cancer; and impact of pre-operative RT on pathology interpretation.

## 3. Pre-Operative RT for Muscle-Invasive Bladder Cancer

The RC associated with pelvic lymph node dissection (PLND) is the most common local therapy in the management of non-metastatic muscle invasive bladder cancer (MIBC) [3]. For eligible patients, neo-adjuvant chemotherapy (NAC) has shown a significant overall survival (OS) benefit in this population. Nevertheless, patients with locally advanced MIBC at the time of RC have an estimated 5-year overall survival of only 10–40% [6]. Loco-regional recurrence (LRR) remains a common and early event, occurring in as many as 40% of the cases in MIBC patients who are treated surgically, something that is associated with the development of distant metastases, and whose therapeutic management generate morbidity and mortality [7,8]. Pathological factors are correlated with LRR; patients with pT3-T4a and/or positive lymph-nodes and/or limited pelvic lymph-node dissection and/or positive surgical margins have been distributed in LRR risk groups with accuracy [9]. There is growing evidence showing that improvements in these intermediate end point translates into benefits in survival for well-selected patients. Peri-operative RT, especially when delivered in the adjuvant setting, has shown the potential to decrease the risk of LRR and improves disease-free survival (DFS) [7,10]. Results from ongoing phase 2 trials evaluating adjuvant RT are awaited [11].

Preoperative RT, offered at a dose and in a clinical target volume different from what is the case in a conservative approach, is a potentially attractive treatment option to prevent intraoperative tumor seeding in the operative field and to sterilize microscopic extension in the perivesical and the removed pelvic lymph nodes. Mostly in the 1970s and 1980s, six clinical trials randomizing preoperative RT versus RC alone were conducted. Five studies did not find any significant improvement with the adjunction of RT in terms of OS or LRR control [12,13,14,15,16]. Awwad et al., in a Middle East population where urothelial carcinoma was not the predominant histology, reported an improvement in the 2-year DFS, but the study was limited to patients with locally advanced disease (cT3-T4 disease) [17]. A first meta-analysis including prospective studies was published in 1998 and showed no benefit for preoperative RT in this population [18]. More recently, in a meta-analysis of seven preoperative radiotherapy trials, a total of 996 patients was analyzed. There was a non-statistically significant improvement in OS for patients who received preoperative irradiation. At three years and five years, the odds ratios were 1.23 (95% confidence interval (CI) 0.72–2.09) and 1.26 (95% CI 0.76–2.09), respectively, in favor of preoperative RT. The authors highlighted a need for current randomized control trials to further evaluate this approach [19].

Whether peri-operative RT is tolerable is a subject of controversy. If early studies from the 1970s were limited by poor patient selection and included of early stage patients who were unlikely to benefit from preoperative irradiation, then they also showed high toxicity rates in the two-dimensional RT era. For example, in a series of patients receiving old-fashioned pre- or post-operative RT, Reisinger et al. observed a 37% rate of late bowel toxicity [20]. The study by El-Monim et al. that compared pre- and post-operative irradiation, giving 50 Gy in conventional fractionation to a pelvic clinical target volume, reported a 6.5% rate of late gastro-intestinal toxicity [21]. Modern RT techniques such as intensity-modulated RT can meaningfully reduce dose to digestive structures, likely improving the risk-benefit profile of this peri-operative strategy [22]. If MIBC has been considered to be poorly sensitive to RT in the past, a RT dose–response effect has been suggested in several series [23,24]. For example, Pos et al. estimated that an increase in the total dose of 10 Gy was associated with a 1.44-fold increase in 3-year local control. Giving a total dose superior to 60 Gy is actually feasible and is recommended in a curative RT intent [25]. The main studies are listed in Table 1.

Preoperative RT has several theoretical advantages over adjuvant RT, including the possibility of treating smaller target volumes and less irradiated normal tissues, better delineation accuracy, and enhanced tumour oxygenation that could improve the effectiveness of irradiation. However, the feasibility of RC following RT can be controversial, and strategies to enhance digestive tolerance may be necessary to reconsider this approach. As evaluated in prostate cancer RT, a rectal spacer could provide a physical barrier between the high dose that is immediately adjacent to the bowel and the rectum. Recently, a study comparing prostate RT with and without spacer has demonstrated with up to 5 years of follow-up, that the use of a rectal spacer was associated with the preservation of bowel quality of life [26]. Similarly, the use of spacers could be evaluated in bladder cancer treated in the preoperative RT setting. Thus, this approach is being reconsidered by some, as improvements in preoperative imaging and radiomics and the emergence of genomic biomarker testing allow us to better select patients who might benefit from this strategy [27]. Moreover, with the recent emerging benefits from neo-adjuvant immunotherapy trials, since recent preclinical and early clinical trials propose a synergistic effect of radiation and immunotherapy, this combination seems to be a new pathway in the treatment of MIBC. Daro-Faye et al. reviewed all of the currently published data discussing the combination of RT and immunotherapy and found that together, this combination result in increasing the immune markers and immunogenic tumor cell death and that it improved tumor control with an acceptable safety profile [28]. Therefore, the potential synergy of RT and immunotherapy to improve LRR and other oncologic outcomes in this otherwise poor prognostic subgroup with locally advanced MIBC must be explored in clinical trials. The currently accruing RACE IT German trial is a single-arm study that is evaluating safety and feasibility of the combined application of preoperative RT with the PD-1 checkpoint-inhibitor Nivolumab followed by RC in patients with locally advanced bladder cancer, and its results are awaited [29]. Finally, preoperative RT must only be proposed in a clinical trial setting.

## 4. Pre-Operative RT for Prostate Cancer

In prostate cancer patients, adjuvant RT following prostatectomy was shown to increase biochemical relapse-free survival [30,31,32]. A not negligible proportion of men who undergo surgery for localized PCa will experience PSA failure that may require post-operative RT. Such a salvage therapy induces late cumulative toxicity and may prevent the rapid recovery of functional outcomes after surgery (mainly urinary continence) when given early after surgery. Thus, there is a need for improving the initial treatment regimen in order to limit the side effects of salvage and unplanned treatment at recurrence. Moreover, despite high-level evidence showing survival advantage, post-operative RT remains under used. The main pros and cons of preoperative RT are highlighted in Figure 1.

The migration towards the use of RT in the preoperative setting has been a success in other malignancies, such as locally advanced rectal cancer and sarcoma [33,34]. In rectal cancer or sarcoma, pre-operative RT is a validated option since it reduces local relapse rates and facilitates conservative surgery [33,34]. Numerous advantages of this regimen have been hypothesized, such as the possibility of tumor downstaging, the reduction in positive surgical margin rates, or the delivery of RT to a smaller volume compared to a post-operative RT regimen. Preoperative RT could also lead to a significant decrease in RT-related toxicity by targeting a gland with a correct prostatic blood supply.

Several authors have therefore hypothesized that pre-operative RT could downstage locally advanced tumors, decrease positive margin rates, and thereby improve survival [35]. The Mayo Clinic retrospectively reported their experience of 18 patients with locally advanced prostate cancer, where pre-operative 2D RT (40–70 Gy) was followed by a radical prostatectomy within 1–2 months [36]. Patients were selected for a radical prostatectomy on the basis of the post-radiation local response assessment of a persisting disease. A minimal postoperative morbidity was reported, and 67% of patients were metastasis free at 5 years. The University of Portland reported a phase I study of pre-operative irradiation (45 Gy in 5 fractions) combined with docetaxel (30 mg/m^2^) in 12 patients with high-risk prostate cancer [37]. The surgical margins were negative in 75% of cases, and the post-operative PSA levels were undetectable in all of the patients at the cost of limited hematological toxicity. Duke University reported a phase 1 study exploring the safety of pre-operative radiotherapy (whole-pelvis up to 54 Gy in 30 fractions) delivered 4–8 weeks prior to radical prostatectomy in 12 patients with high-risk prostate cancer [38]. No intraoperative morbidity was reported, but two patients developed a symptomatic grade 3 urethral stricture. The University of Toronto also conducted a phase 1 trial of ultra-hypofractionated pre-operative radiotherapy (25 Gy in 5 fractions) followed by surgery 1–2 weeks later in 13 patients [39]. Limited peri-operative complications were observed, with only one patient having signs of intra-operative inflammation. Late severe urinary incontinence was noted in 2 out of 13 patients. Updated data after a median follow-up of 12.2 years showed that long-term urinary toxicity rates were high, with two and three patients experiencing grade 2 or 3 toxicity [40]. Similarly, the University of California Los Angeles reported a phase 1 trial of ultra-hypofractionated RT (24 Gy in 3 fractions) prior to radical prostatectomy in 11 patients with high-risk prostate cancer [41]. Acute and late grade 3 urinary toxicity was reported in two patients with incontinence. All of the main studies are reported in Table 2.

Researchers are actively pursuing their efforts with at least three ongoing clinical trials evaluating ultra-hypofractionated pre-operative irradiation (NCT02572284; NCT02946008; NCT03663218). Altogether, these studies suggest that pre-operative RT is feasible with moderate acute urinary morbidity but with possibly higher rates of severe urinary incontinence than with RP alone, especially when using ultra-hypofractionated RT. None of the studies reported unusual rates of rectal toxicity. Oncologic outcomes are difficult to evaluate in these small series of patients with high-risk tumors but do not seem to be substantially different from patients who are undergoing postoperative RT.

Prostate cancer evolution is definitively different from the natural history of other malignancies that are often treated by pre-operative RT, such as rectal cancer. However, the benefit of local intensification by RT may have a significant impact in a well-selected patient sub-population. Such a selection may be driven by usual clinico-pathological factors but also novel biomarkers aiming to identify patients who are at a very high-risk of local relapse. Imaging might play an important role in improving the selection of the best candidates for intensive, combined local treatment strategies. Multiparametric MRI and PSMA positron emission tomography have demonstrated their usefulness for identifying adverse pathologic features, such as seminal vesicle invasion or extraprostatic extension, and for detecting pelvic and distant metastases that are non-visible on conventional imaging. The routine use of these techniques could enhance precise tumor delineation and tailored preoperative RT-based strategies thanks to a better anatomical perception of the disease.

Currently, although preoperative RT is increasingly used in other malignancies, serious limitations remain for PCa management, and preliminary findings from clinical trials should be considered cautiously, mainly due to a high rate of clinically significant late toxicities.

Therefore, preoperative RT for PCa should remain evaluated in clinical trials. Further studies should determine the optimal dose schedule and delivery techniques more precisely in the era of hypofractionated regimens and of imaging-guided therapies. New technologies such as stereotactic body RT may play a key role in the near future for improving precise planning and in providing an optimal boost to the index lesion.

## 5. Impact of Pre-Operative Radiotherapy on Imaging Interpretation

The role of imaging is to identify patients with pure local recurrence and to guide prostate biopsies in order to prepare for salvage treatments by surgery or focal therapy. There is no indication that it is necessary to perform imaging outside of the context of biochemical or clinical recurrence. Currently, unlike in the BCR post PR setting, mpMRI is a well validated tool for patients without distant metastases and is fit for local salvage therapy [4,42]. So far, the EAU has recommended mpMRI as the best technique to assess local recurrence and to guide targeted biopsies for patients who are considered for local salvage therapy [4]. The acquisition protocol is the same as it is for prostate cancer detection when treating a naïve patient. There is no reporting standardization; indeed, the PI-RADS score was designed for naïve prostate glands [43]. Suspicious lesions should be graded on a Likert scale, and the maximal diameter and location must be specified.

RT induces changes in the prostate, including gland shrinkage, loss of normal anatomy, and decreased contrast between PCa and normal prostatic tissues on T2-weighted (T2W) imaging due to glandular atrophy and fibrosis. Thus, recurrence can be difficult to detect on T2-weighetd images, and the use of functional sequences seems essential (Figure 2; [44,45]).

Diffusion and dynamic contrast-enhanced imaging have been shown to accurately identify local recurrence in the irradiated prostate [46,47]. DW imaging is particularly efficient in this indication; the sensitivity and specificity of the association T2-weighted and DWI imaging are, respectively, 94% and 75%, for the detection of recurrences > 0.4 cm^2^ [48].

In the field of muscle-invasive bladder cancer, no data on RT impact on imaging are available. However, future studies will certainly emerge, given the development of bladder-sparing strategies such as trimodality treatment and the increased use and standardization of pelvic MRI for staging purposes and for assessing the efficacy of bladder preservation therapies [49].

## 6. Radical Surgery after Pre-Operative Radiotherapy

### 6.1. Radical Cystectomy (RC)

The choice of bladder-sparing techniques aims to avoid radical surgery. However, in some cases (20–30%), salvage radical cystectomy (sRC) remains mandatory in cases of local disease persistence or progression after RT initiation. The performance of sRC can be challenging and can more frequently have an impact on complications and functional outcomes compared to upfront RC.

One of the most recent series found that the risk of sRC at 5 years was 29% [50]. A recent systematic review of the literature gathered 73 studies comprising 9110 patients undergoing trimodality therapy (TMT) and aimed to identify the risk of sRC and its outcomes [51]. The pooled rate of sRC was 19.2% for studies with at least a mid-term follow-up. The main reasons for sRC were non-response to TMT and local recurrence. Thus, the proportion of early and late sRC was comparable (55.7% versus 44.3%).

Whereas the proportion of continence diversion and orthotopic neobladder continuously increased in a contemporary series of primary RC, such diversions are challenging in the context of previous pelvic RT. In the systematic review from Schuettfort et al., the pooled rate of incontinent urinary diversion was 91% [51]. In the series from Chahal et al., only 3.5% of sRC patients had a continent urinary diversion [52]. Few studies have assessed the complication rates of sRC, with an overall rate ranging from 65% to 72%. The overall 30-day mortality was estimated at 0–8.8%. These results are not far off from the rates of mortality and morbidity of radical cystectomy in the primary setting for patients who are not irradiated.

In a series of 91 patients undergoing sRC after TMT, Eswara et al. reported 99% of ileal conduit diversion and 69% of perioperative complications [53]. The rate of major complications was 16% with a 21%-risk of readmission during the first 3 months after sRC. The overall mortality rate was 2.2%. Significant cardiovascular and tissue healing complications such as fascial dehiscence, wound infection, and ureteral stricture were more frequently reported after sRC compared to after immediate RC (35% vs. 12%, *p* = 0.05).

The Massachussetts General Hospital team recently updated the outcomes for patients undergoing sRC after TMT with those undergoing primary RC [54]. Overall, 265 patients were retrospectively included during a 10-year period (median follow-up: 65 months). They did not find any differences between groups in terms of intraoperative and early complications. However, sRC was correlated, with more frequent late complications having a higher incidence than any late (HR 2.3, *p* = 0.02) and major late complications (HR 2.1, *p* < 0.05). The survival outcomes were not impacted by the use of pre-operative RT, as no difference was observed in terms of disease-specific and overall survival. Even if the 30-day mortality rates for sRC appeared to be low, non-randomized comparisons highlighted a higher risk compared to patients who had been treated by primary RC. With respect to specific complications, some series have reported higher rates of overall urinary anastomosis-related complications and major gastrointestinal complications such as bowel leakage following sRC when compared to primary RC [52,53,54,55].

Globally, although no direct, prospective comparison is available, patients undergoing sRC might lose opportunities for a high-quality upfront RC in terms of nerve-sparing techniques, perioperative outcomes, and urinary diversion. An important selection bias must be taken into consideration regarding the feasibility of sRC in the patients included in analyzed studies. The impact of late complications on reoperation needs and on quality-of-life also warrants further evaluation.

### 6.2. Radical Prostatectomy (RP)

Surgery is a valid treatment option for local recurrence management after RT. Local recurrence often occurs several months after RT. Thus, no data exists regarding the safety of immediate sRP after pre-operative RT. However, several retrospective studies have assessed the functional outcomes after sRP in the context of delayed surgery for local recurrence. The overall median time to sRP was 51 months in a recent literature review [56]. Globally, sRP appeared to be feasible without a significant increase in major complication rates. Pre-operative RT might also have an impact on the lymph node dissection realization in the context of neoadjuvant pelvic irradiation. sRP was mostly performed using an open approach (80% of the patients). However, the use of robotic has continued to increase over the last decade. The main issue of this salvage procedure continues to be a high rate of urinary incontinence. The rates of urinary continence varied from 20% to 90% at 1 year in the literature, according to the definition that was used (number of daily pads) [56].

Nerve-sparing techniques cannot be performed for the majority of patients because of the need for extensive surgery in the context of local recurrence and given the perioperative technical difficulties that are caused by the peri-prostatic fibrosis that is induced by a previous RT regimen. Grubmüller et al. reported a post-sRP erectile function rate ranging from 0% to 13% [56]. In a recent, comparative series of robotic sRP versus primary RP, Nathan et al. did not find significant differences in terms of high-grade complication rates [57]. sRP was associated with lower rates of continence recovery (78.8% versus 84.3% at 2 years; *p* = 0.337) and higher percentages of erectile dysfunction (94.8% versus 76.3%; *p* < 0.001).

## 7. Impact of Pre-Operative Radiotherapy on Pathology Interpretation

The analysis of prostate tissues that have been previously treated by RT could be challenging because of the histopathological changes that are induced in both the benign prostate and in the cancer foci. The difficulties include the distinction of treatment effects in non-tumor prostate tissue from residual tumors.

In non-tumor prostate tissues, radiation can induce vascular changes, such as the narrowing of the lumen and myo-intimal proliferation, together with stromal fibrosis. RT can also induce atrophia and cytologic atypia of degenerative nature in benign glands that could persist for a long time after the initial treatment [58,59].

In residual tumor areas, the effect of RT is variable, from minimal changes to pronounced modifications [60]. After RT, the neoplastic glands become smaller, the nuclei are pycnotic or large, and the nucleoli (key criteria of malignancy for prostate cancer diagnosis) is often lost. Moreover, RT could also induce architectural changes in the tumor, resulting in the loss of the glandular pattern [59,60].

While the residual tumor can be identified using routine staining in most cases, immunohistochemistry may sometimes be required. Fortunately, the expression of racemase by cancer cells is not altered by RT, and p63, a marker of basal cells, remains negative, regardless of treatment duration [61]. Therefore, the use of the cocktail p63/racemase could be used to distinguish between residual tumor and RT-induced atypia.

Since RT could induce changes in the tumor architecture, this could therefore lead to spurious increases in ISUP grading. A scoring system has been proposed to evaluate the degree of radiation changes, and it has been suggested that cancer foci showing moderate to severe treatment effects should not be histologically graded according to the ISUP classification [62]. In contrast, the ISUP grade group could be given in those cases where there are minimal RT-associated changes [59,60].

## 8. Conclusions

Radical surgery and RT are mainly used as competitive treatments with curative intent in uro-oncology. In localized prostate and bladder cancer, RT is preferentially proposed as an initial single-modality local therapy or as a salvage local recurrence management strategy after surgery. However, preliminary studies suggest that both modalities can be complementary and that a pre-operative radiotherapy strategy could be beneficial in a well-defined population of patients who have been determined to be at a very high-risk of local relapse. Future prospective trials are warranted to evaluate the oncologic benefit of such a combination of local treatments in addition to new life-prolonging systemic therapies such as immunotherapy and new generation hormone therapies. Moreover, the safety and the feasibility of salvage surgical procedures remain poorly evaluated in that context due to non-response or local recurrence after pelvic RT.

Thus, preoperative RT in uro-oncology should continue to be evaluated in clinical trials. Further studies should determine the optimal dose schedule and delivery techniques more precisely in the era of hypofractionated regimens and of imaging-guided therapies. New technologies such as stereotactic body RT may play a key role in the near future for improving precise planning and optimal boost to the index lesion.

## Figures and Tables

**Figure 1 cancers-13-06070-f001:**
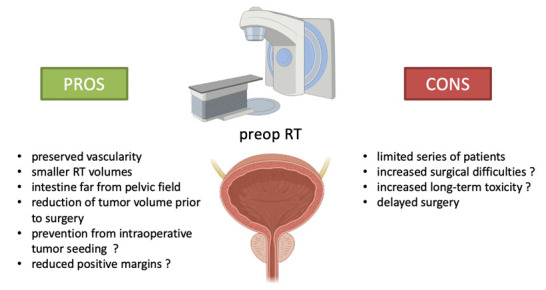
Preoperative RT: pros and cons.

**Figure 2 cancers-13-06070-f002:**
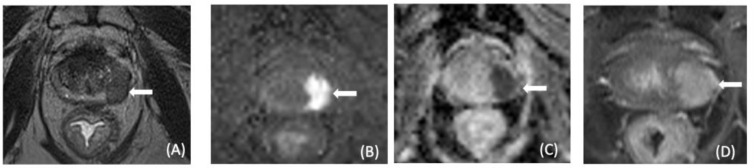
A 77-year-old male with serum PSA = 7 ng/mL after RTE. Axial T2 W MRI (**A**), DW MR(**B**), ADC map of DW MR (**C**), and DCE MRI (**D**) show a lesion in the left part of the peripheral zone (arrows). Targeted biopsy revealed Gleason 3 + 4 recurrent prostate cancer.

**Table 1 cancers-13-06070-t001:** Summary of pre-operative radiotherapy vs. control studies in bladder cancer patients. RT: radiotherapy; f: fractions; pts: patients.

Reference	pts	Treatment	Dose	Time Interval between RT and Surgery	Outcomes
Ghoneim et al. [12]	92	2D Pelvic lymph-nodes RT	20 Gy in 5 f	3 days	3-year Overall Survival: 0.52 vs. 0.48 (NS except for locally advanced and high grade tumors)
Blackard et al. [13]	45	2D Bladder RT	45 Gy-	4–6 weeks	3-year Overall Survival: 0.4 vs. 0.4
Slack et al. [14]	229	2D Entire pelvis RT	45 Gy-	4–8 weeks	3-year Overall Survival: 0.5 vs. 0.37
Anderstrom et al. [15]	44	2D Entire pelvis	32–54 Gy/20–30 f/4–6 wk	2–4 weeks	3-years Overall Survival: 0.81 vs. 0.81 Tumor eradication achieved in 80% of patients when high doses were administered
Smith et al. [16]	124	2D Pelvic lymph-nodes RT	20 Gy in 5 f	1 week	3-year Overall Survival: 0.65 vs. 0.48 (NS)
Awwad et al. [17]	48	Entire pelvis	-Split course arm: 20 Gy/10 f for 1 wk × 2 (1 week break between) -Hyperfractionation arm: 20 Gy/34 f for 2 d × 2 (1 week break between)	2–3 weeks	2-year Disease-free survival rate: 53 +/− 9% vs. 19 +/− 10% Post-irradiation tumor shrinkage was noted in the majority of patients

Overall survival values indicate the proportion of patients alive in each study arm at given time point. Disease-free survival values indicate the proportion of patients alive in each study arm at given time point. −: not reported. NS = non-significant.

**Table 2 cancers-13-06070-t002:** Summary of pre-operative radiotherapy studies in locally advanced and high-risk prostate cancer patients. RT: radiotherapy; f: fractions; pts: patients.

Institution	pts	Treatment	Dose	Time Interval between RT and Surgery	Tolerance	Outcomes
Mayo Clinic	18	2D prostate RT	40–70 Gy	1–2 months	Minimal postoperative morbidity	Metastasis-free survival at 5 years: 67%
University of Portland	12	Prostate RT combined with docetaxel (30 mg/m^2^)	45 Gy in 5 f		Limited hematological toxicity	Surgical margins negative in 75%; post-operative PSA levels undetectable in all patients
Duke University	12	Whole pelvis RT	54 Gy in 30 f	4–8 weeks	No intraoperative morbidity; grade 3 urethral stricture in pts	Two-year actuarial biochemical recurrence-free survival 67%
University of Toronto	13	Ultra-hypofractionated prostate RT	25 Gy in 5 f	1–2 weeks	Signs of intra-operative inflammation in 1 pt; Late grade 3 urinary toxicity in 3 pts	Biochemical relapse-free survival at 3 years 83%
University of California Los Angeles	11	Ultra-hypofractionated prostate RT; additional nodal RT with androgen suppression in 2 pN1 pts	24 Gy in 3 f		Acute and late grade 3 incontinence in 2 pts	Biochemical recurrence within the first 12 month in 2 pts

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
