# Peer review of "Oncologic Impact and Safety of Pre-Operative Radiotherapy in Localized Prostate and Bladder Cancer: A Comprehensive Review from the Cancerology Committee of the Association Française d’Urologie"

_cancers, 2021, doi:10.3390/cancers13236070_

Round 1

Reviewer 1 Report

In this manuscript, the authors reviewed the impact and safety of preoperative Radiotherapy in prostate and bladder cancer. This review provides us the information which can be decided to choose the preoperative radiotherapy. The topic is not new but it is valuable to summary all experiences or studies. However, it was written too simple or not so deep. Especially,  the authors didn't summary about what situation it can be used and what is its advantages and disadvantages compared with postoperative radiotherapy. The whole logic is not written so smoothly or not so interesting for readers. If the authors can draw some figures or tables to enhance these issues, it will benefit a lot.

Author Response

Comment 1

In this manuscript, the authors reviewed the impact and safety of preoperative Radiotherapy in prostate and bladder cancer. This review provides us the information which can be decided to choose the preoperative radiotherapy. The topic is not new but it is valuable to summary all experiences or studies. However, it was written too simple or not so deep. Especially,  the authors didn't summary about what situation it can be used and what is its advantages and disadvantages compared with postoperative radiotherapy. The whole logic is not written so smoothly or not so interesting for readers. If the authors can draw some figures or tables to enhance these issues, it will benefit a lot.

We would like to thank the reviewer for this comment. We have had a dedicated part in the chapter 3 explaining the potential advantages/disadvantages for the preoperative approach and some data focusing on spacers showing how to improve tolerance. Then, we have specified lastly that this approach must be proposed only in clinical trials. We also have added a table with preoperative studies and their results. We added a figure summarizing the pros and cons of preop RT strategies.

Reviewer 2 Report

Some minor issues need to be corrected in the revised version:

  1. line 208-211: this sentance seems have some problem of grammer issue;
  2. In the text, if you give the abbreviation of RT, you should use it throughout; the samething RC;
  3. line 220: there is a typo, sucha (should be such);

This is very intersting review article in the field of prostate and bladder cancer area and that would be a graet help to the professional doctors, as well as the basic reaserch PI. 

Author Response

Comment 2

Some minor issues need to be corrected in the revised version:

  1. line 208-211: this sentance seems have some problem of grammer issue;
  2. In the text, if you give the abbreviation of RT, you should use it throughout; the samething RC;
  3. line 220: there is a typo, sucha (should be such);

This is very intersting review article in the field of prostate and bladder cancer area and that would be a graet help to the professional doctors, as well as the basic reaserch PI. 

We would like to thank the reviewer for this elegant comment. We have modified the manuscript following your suggestions, correcting typo and phrasing.

Reviewer 3 Report

General comments

This paper reviewed the potential usefulness of preoperative radiotherapy in prostate and muscle-invasive bladder cancer, aiming to enhance pathological response and local control, and to prevent from intraoperative tumor seeding. There are minor concerns that I would suggest.

Specific comments

Minor

1) Pre-operative radiotherapy studies in prostate cancer were shown in Table 1, I recommend the authors to make summary table for bladder cancer studies.

2) Rectal injury is the most critical complication during operation after preoperative radiotherapy. Recently, SpaceOAR Hydrogel is available. The authors should propose any ideas to reduce complications during operation after preoperative radiotherapy.

Author Response

Comment 3

General comments

This paper reviewed the potential usefulness of preoperative radiotherapy in prostate and muscle-invasive bladder cancer, aiming to enhance pathological response and local control, and to prevent from intraoperative tumor seeding. There are minor concerns that I would suggest.

Specific comments

Minor

  • Pre-operative radiotherapy studies in prostate cancer were shown in Table 1, I recommend the authors to make summary table for bladder cancer studies.

Thanks for this comment.

A table has been added to present preoperative RT studies in bladder cancer.

  • Rectal injury is the most critical complication during operation after preoperative radiotherapy. Recently, SpaceOAR Hydrogel is available. The authors should propose any ideas to reduce complications during operation after preoperative radiotherapy.

Thanks for this comment. Following our sentences explaining that technical implementation of IMRT and IGRT could be a way to reduce toxicity following RT, we have highlighted data on spacers with potential benefits showed in patients QOL.

Reviewer 4 Report

This study was reported the oncological impact and safety of preoperative RT for MIBC and prostate cancer. The reviewer would like to suggest some critiques as follows.

  1. The reviewer thinks that urothelial carcinoma cells are low sensitive to RT. On line 139, “Preoperative RT…..theoretical advantages” is unclear.
  2. Several guidelines are not recommended RT for MIBC. Therefore, the reviewer thinks that RT for MIBC should be denied.
  3. On line 114, the reviewer thinks that RT is one of the local treatments. “Preoperative….lymph nodes” should be revise in theory.
  4. The reviewer thinks that RT for prostate cancer is effective treatment. However, the utility of neoadjuvant RT followed by surgery is controversial. The authors described several reports regarding preoperative RT with prostatectomy. However, the enrolled patients in these studies were very small. Preoperative RT for prostatectomy is skeptical.

Author Response

Comment 4

This study was reported the oncological impact and safety of preoperative RT for MIBC and prostate cancer. The reviewer would like to suggest some critiques as follows.

  1. The reviewer thinks that urothelial carcinoma cells are low sensitive to RT. On line 139, “Preoperative RT…..theoretical advantages” is unclear.

Thanks. We have proposed to cite some key papers highlighting the dose-response effect observed in RT for bladder cancers.

  1. Several guidelines are not recommended RT for MIBC. Therefore, the reviewer thinks that RT for MIBC should be denied.

Thanks for this comment. In the EAU, AFU and NCCN guidelines, RT as part of trimodal therapy is recommended for well selected patients.

  1. On line 114, the reviewer thinks that RT is one of the local treatments. “Preoperative….lymph nodes” should be revise in theory.

We have modified the sentence in this direction, mitigating the message. Thank you.

  1. The reviewer thinks that RT for prostate cancer is effective treatment. However, the utility of neoadjuvant RT followed by surgery is controversial. The authors described several reports regarding preoperative RT with prostatectomy. However, the enrolled patients in these studies were very small. Preoperative RT for prostatectomy is skeptical.

We fully agree. Although provocative, these small series of patients suggest that preop RT strategies in prostate cancer may lead to increased long-term toxicity while no real synergistic action with surgery was shown. As stated in the text “preoperative RT for PCa should remain evaluated in clinical trials. Further studies should determine more precisely the optimal dose schedule and delivery techniques in the era of hypofractionated regimens and of imaging-guided therapies. »

Round 2

Reviewer 4 Report

The authors revised this manuscript according to the reviewer’s recommendation.  The reviewer believes that this paper will provide useful information for the readers.